# L-Thyroxine Improves Vestibular Compensation in a Rat Model of Acute Peripheral Vestibulopathy: Cellular and Behavioral Aspects

**DOI:** 10.3390/cells11040684

**Published:** 2022-02-16

**Authors:** Guillaume Rastoldo, Emna Marouane, Nada El-Mahmoudi, David Péricat, Isabelle Watabe, Agnes Lapotre, Alain Tonetto, Alejandra López-Juárez, Abdessadek El-Ahmadi, Philippe Caron, Marie-José Esteve Fraysse, Christian Chabbert, Andreas Zwergal, Brahim Tighilet

**Affiliations:** 1Aix Marseille Université-CNRS, Laboratoire de Neurosciences Cognitives, LNC UMR 7291, 13331 Marseille, France; guillaume.rastoldo@univ-amu.fr (G.R.); emna.marouane@gmail.com (E.M.); nada.el-mahmoudi@univ-amu.fr (N.E.-M.); isabelle.watabe@univ-amu.fr (I.W.); agnes.lapotre@univ-amu.fr (A.L.); abdessadek.el-ahmadi@univ-amu.fr (A.E.-A.); christian.chabbert@univ-amu.fr (C.C.); 2Institute of Pharmacology and Structural Biology (IPBS), University of Toulouse, CNRS, 31400 Toulouse, France; david.pericat@ipbs.fr; 3Centrale Marseille, Aix Marseille Université-CNRS, FSCM (FR1739), PRATIM, 13331 Marseille, France; alain.tonetto@univ-amu.fr; 4Department of Chemical, Electronic and Biomedical Engineering, Division of Sciences and Engineering, University of Guanajuato, Guanajuato 38116, Mexico; lopez.sa@ugto.mx; 5Service d’Endocrinologie et Maladies Métaboliques, Pôle Cardio-Vasculaire et Métabolique, CHU de Rangueil-Larrey, 31059 Toulouse, France; caron.philippe.j@orange.fr; 6Service d’Oto-Rhino-Laryngologie, d’Oto-Neurologie et d’ORL Pédiatrique, Centre Hospitalier Universitaire de Toulouse, 31400 Toulouse, France; fraysse.mj@chu-toulouse.fr; 7GDR Physiopathologie Vestibulaire—Unité GDR2074, CNRS, 13003 Marseille, France; 8Department of Neurology, University Hospital, LMU Munich, 80539 Munich, Germany; andreas.zwergal@med.uni-muenchen.de; 9German Center for Vertigo and Balance Disorders, DSGZ, LMU Munich, 80539 Munich, Germany

**Keywords:** vestibular compensation, thyroid hormones, vertigo pharmacology, neurogenesis, microglial reaction, brain metabolism, vestibular nuclei

## Abstract

Unilateral vestibular lesions induce a vestibular syndrome, which recovers over time due to vestibular compensation. The therapeutic effect of L-Thyroxine (L-T4) on vestibular compensation was investigated by behavioral testing and immunohistochemical analysis in a rat model of unilateral vestibular neurectomy (UVN). We demonstrated that a short-term L-T4 treatment reduced the vestibular syndrome and significantly promoted vestibular compensation. Thyroid hormone receptors (TRα and TRβ) and type II iodothyronine deiodinase (DIO2) were present in the vestibular nuclei (VN), supporting a local action of L-T4. We confirmed the T4-induced metabolic effects by demonstrating an increase in the number of cytochrome oxidase-labeled neurons in the VN three days after the lesion. L-T4 treatment modulated glial reaction by decreasing both microglia and oligodendrocytes in the deafferented VN three days after UVN and increased cell proliferation. Survival of newly generated cells in the deafferented vestibular nuclei was not affected, but microglial rather than neuronal differentiation was favored by L-T4 treatment.

## 1. Introduction

A unilateral vestibular lesion induces in most species, including humans, a characteristic vestibular syndrome composed of oculomotor, posturo-locomotor, and perceptive–cognitive deficits. These symptoms recover over days to weeks due to a well-known phenomenon called vestibular compensation. In neuroscience, this functional recovery is considered as a good example of postlesional plasticity. Vestibular compensation is based on various plasticity mechanisms to restore a balanced activity between bilateral vestibular complexes in the brainstem [1,2,3].

Pharmacological modulation of the level of excitability or neurogliogenesis in the deafferented vestibular nuclei (VN) modifies the kinetics of vestibular functional restoration. Indeed, the pharmacological blocking of SK channels (small conductance calcium-activated potassium) by apamin, which inhibits the hyperpolarization phase of the action potential, facilitates the restoration of posturo-locomotor function and gaze stabilization [4]. Among the numerous mechanisms of plasticity promoting vestibular compensation, we have demonstrated that a unilateral vestibular neurectomy (UVN) induces a reactive neurogenesis in the deafferented vestibular nuclei of adult cats and rats [5,6,7,8,9,10]. Continuous intracerebroventricular administration of an antimitotic drug immediately after UVN completely blocks cellular proliferation and neurogliogenesis and delays the posturo-locomotor function recovery drastically [5]. Conversely, increasing cell proliferation and survival with a continuous infusion of brain derived neurotrophic factor (BDNF) immediately after UVN accelerates the recovery of balance and posture, while blocking BDNF-TrkB signaling significantly reduces cell proliferation and prevents postural locomotor recovery in animals with vestibular injuries [8]. These data indicated that the priority of the deafferented vestibular environment after UVN is to promote both the restoration of homeostatic excitability between homologous VN and the expression of neurogliogenesis for vestibular functional recovery.

Interestingly, thyroid hormones (TH) can promote these two important mechanisms, which are essential for vestibular functional recovery. TH signaling governs many aspects of neurogenesis, including proliferation, survival, migration, differentiation, and maturation of neuronal and glial cells [11,12,13,14]. Thyroxine (T4) or triiodothyronine (T3) injections have been shown to increase the expression of BDNF in different pathological models (stroke [15,16], Alzheimer’s disease [17], and axotomy [18]). In addition, several studies have demonstrated that TH treatment after traumatic brain injury reduces lesion size, inflammation, and promotes neurogenesis and neuronal survival to improve functional recovery through genomic and non-genomic actions (for review see [19,20]). Finally, even though the molecular events by which TH regulate energy metabolism are still not fully understood [21,22,23], this essential function could rebalance the activity between the two homologous VN and promote vestibular compensation. Based on all this evidence, TH are likely candidates to alleviate acute vestibular syndrome and facilitate vestibular compensation.

The aim of our study was to determine whether short-term pharmacological treatment with L-thyroxine following UVN in adult rats affects the time course of vestibular functional recovery and post-lesion plasticity mechanisms in the deafferented VN.

## 2. Materials and Methods

### 2.1. Animals

The experiments were performed on 28 adult Long Evans rats (250/300 g) originating from our own breeding, from parents obtained from Charles River (St Germain sur l’Arbresle, France). All experiments were performed in accordance with the European Union 2010/63/EU Directive and under the veterinary supervision and control of the National Ethical Committee (French Agriculture Ministry Authorization: B13-055-25). The present study was specifically approved by the Neurosciences Ethic Committee N°71 of the French National Committee of animal experimentation. Every attempt was made to minimize both the number and the suffering of animals used in this experiment. The animals were housed in a large, confined space with 12 h–12 h diurnal light variations and free access to food and water.

### 2.2. Unilateral Vestibular Neurectomy

Animals were subjected to a left-side vestibular nerve section (*n* = 24), following the surgical procedure previously reported in the literature [24]. Thirty minutes after a subcutaneous injection of buprenorphine (Buprecare^®^; 0.02 mg/kg), the rats were placed in the anesthesia induction box and left for 5 min (isoflurane concentration 4%). Once they were deeply anesthetized, they were intubated and, during the surgery, the anesthesia was maintained at an isoflurane concentration of 3%. A tympanic bulla approach gave access to the vestibular nerve: the cervical muscular planes were dissected leading to the tympanic bulla, which was widely drilled to expose the stapedial artery and the promontory containing the cochlea. The cochlea was drilled, exposing the cochlear nerve. The cochlear nerve meatus was enlarged with a needle leading to the vestibulocochlear nerve, which was sectioned at its entry into the brainstem after aspiration of Scarpa’s ganglion. The wound was closed using a stapler. Before awakening, the animals received either an intraperitoneal injection of 0.9% NaCl (UVN-NaCl group) or a solution of thyroxine (UVN-T4 group, 10 µg/kg). A solution of Ringer Lactate (Virbac, Carros, France; 10 mL/kg) was also administered subcutaneously in order to alleviate the dehydration resulting from the inability of the animal to drink normally as a result of the injury.

The success of the surgery was attested at the behavioral level by the presence of a characteristic vestibular syndrome and at the histological level by the observation of the full section of the 8th cranial nerve between Scarpa’s ganglion and VN in the brainstem under optical microscopy (see [24] for details).

### 2.3. Criteria for Exclusion

Animals were excluded from the study if the following symptoms were observed:-loss of body weight equal to more than 20% of the pre-operative value.-if the facial nerve had been sectioned.-abnormalities in behavioral scoring, i.e., inability of the animal to stand on all four paws after 5 days post-UVN, convulsions, hemiataxia, etc.

Based on these criteria, one animal of the UVN-T4 group was excluded.

### 2.4. Study Design

Rats were randomly assigned to the following groups:

Non-UVN rats: Control rats were perfused transcardially with saline followed by 4% paraformaldehyde for pre-operative immunohistochemistry assessment (*n* = 4).

UVN-NaCl group: Rats were subjected to UVN, and saline was administered (0.3 mL, i.p.) at the end of the surgery and one injection/day during the first 3 days post-lesion (*n* = 12).

UVN-T4 group: Rats were subjected to UVN and L-Thyroxine (Sigma-Aldrich St. Quentin Fallavier, France, Cat#T2501) was administered (10 μg/kg, i.p.) at both the end of the surgery and one injection/day during the first 3 days post-lesion (*n* = 12). The dose and the administration route used in this study was based on [15].

A part of the UVN-NaCl group and UVN-T4 group was killed at day 3 post-UVN (*n* = 4/group) or day 30 (*n* = 8/group). UVN-NaCl (*n* = 8) and UVN-T4 (*n* = 8) rats underwent behavioral investigation prior to UVN and on days 1, 2, 3, 7, 10, 14, 21, and 30 post-lesion to evaluate the time course of vestibular functional recovery, received a BrdU injection (200 mg/kg, i.p.) on day 3 post-UVN, and were killed at day 30 for cellular investigation. Blinded L-T4 or vehicle injection was given 30 min before the first behavioral test. UVN-NaCl and UVN-T4 groups first performed video-tracking in the open-field, then weight distribution (Bioseb device), and, finally, support surface area. The behavioral tests were conducted 5 min apart.

To study cell proliferation and short-term effect of L-T4 injections UVN-NaCl (*n* = 4) and UVN-T4 (*n* = 4) rats received a BrdU injection (200 mg/kg, i.p.) on day 3 post-UVN and were killed 3 h later.

### 2.5. Qualitative Assessment of the Vestibular Syndrome

We assessed, qualitatively, the vestibular syndrome following UVN. Behavioral symptoms of vestibular imbalance were scored for 10 components after UVN: the tail-hang test, rearing, grooming, displacement, head-tilt, barrel rolling, retropulsion, circling, and bobbing. The tail-hang test (adapted from [25,26,27]), quantifies typical vestibulo-spinal reflexes.

-Tail hanging behavior: Animals were picked up from the ground at the base of the tail and body rotation was scored from 0 point (no rotation) to 3 points (several rotations of 360°)-Landing reflex: After animals were picked up from the ground at the base of the tail, we scored the first 3 landings from 0 (presence of a landing reflex on the 3 landings) to 3 points (absence of landing reflex on the 3 landings). When lifted by the tail, control rats exhibit a landing reflex, consisting of forelimb extension, that allows them to land successfully (i.e., they land on all four legs). Rats with impaired vestibular function do not exhibit a forelimb extension, they spin or bend ventrally, sometimes “crawling” up toward their tails, causing them to miss their landings.-Rearing: the ability of the rat to rear was scored from 0 point (rearing is observed) to 1 point (rearing is absent)-Grooming: the ability of the rat to groom correctly were scored as follows: 0 point (correct grooming of full body) 1 point (grooming of the face, belly, and flanks but not the base of the tail), 2 points (grooming of the face and belly), 3 points (grooming of the face), 4 points (inability of the animal to groom itself)-Displacement: quality of the displacement of the rat was scored from 0 (displacement of the rat with no visible deficit) to 3 points (several deficits in the displacement of the rat)-Head tilt was scored by estimating the angle between the jaw plane and the horizontal with 0 points (absence of a head-tilt) to 3 points (for a 90° angle)-Barrel rolling was scored as follows: 0 points (absence of barrel rolling), 1 point (barrel rolling evoked by an acceleration in the vertical axis of the rat in our hand), 2 points (spontaneous barrel rolling)-Retropulsion characterizes backwards movements and was scored from 0 (absence of retropulsion) to 1 point (presence of retropulsion)-Circling was scored from 0 point (absence of circling behavior) to 1 point (presence of circling behavior)-Bobbing is related to rapid head tilts to the side and was scored from 0 point (absence of bobbing) to 1 point (presence of bobbing)

A first acquisition was done the day before the lesion, serving as a reference value, and then acquisitions were performed at days (D) 1, 2, 3, 7, 10, 14, 21, and 30 post-lesion.

### 2.6. Weight Distribution

To quantify the postural syndrome following UVN we used a device (DWB1^®^, Bioseb, Vitrolles, France) measuring the weight distribution at all contact points of the animal’s body with the ground. This apparatus has previously been described for the assessment of postural instabilities in the same model of vestibular loss [28]. The device consists of a Plexiglas cage (25 × 25 cm) in which the animal can move freely. The floor of this cage is fully covered with a plate with 2000 force sensors. Sensors detect vertical pressure at a sample rate of 30 Hertz. The sensors are connected to an electronic interface that converts the current flowing through it into a measure of weight, the whole being connected to a computer. The cage is closed by a lid on which is attached a high definition camera, also connected to the computer through a USB cable.

The analyses enabled the acquisition of a range of parameters calculated automatically by the software (for details, see [28]). Based on previous results [28], we chose to analyze only the weight distributed on the lateral axis, which allows for the determination of how the animal distributes its weight between the left and right paws in order to maintain balance.

For each acquisition session, the rat was placed in the device for 5 min and could move freely. The pre-operative session was recorded the day before the surgery, and then the time course of the vestibular syndrome was studied on D1, 2, 3, 7, 10, 14, 21, and 30 post-lesion.

### 2.7. Open Field Test

To quantify the locomotor syndrome following UVN, we used an automated video tracking software (EthoVision™ XT 14, Noldus, Wageningen, The Netherlands). Animals were individually placed in a square open field (80 × 80 × 40 cm) and were allowed to move freely for 10 min. Their behavior was recorded for 10 min using a digital camera and analyzed with EthoVision™ XT 14 software. Position of nose, body center, and tail were automatically detected by video software.

Based on previous results [29], the mean distance travelled, meander, mean locomotor velocity, mean number of high accelerations (>50 cm/s^2^), and mean percentage of time spent immobile were quantified (for details, see [29]). To minimize stress, the room was lit as dimly as possible while allowing us to clearly discern the rats. At the beginning of the session, the rat was placed on the right side of the field, head facing the wall. A first acquisition was done the day before the lesion, serving as a reference value, and then acquisitions were performed at D1, 2, 3, 7, 10, 14, 21, and 30 post-lesion.

### 2.8. Support Surface

Static postural deficits and recovery were evaluated by measuring the support surface delimited by the four legs of the rat. The support surface can be regarded as a good estimate of postural control because it reflects the behavioral adaptation of the rat in compensating for the static vestibulospinal deficits induced by the vestibular lesion. To quantify the support surface, rats were placed in a device with a graduated transparent floor that allowed them to be filmed from underneath. A scale drawn on the bottom served to take measurements of the location of the four paws. To avoid measures when the animals were moving, we measured the support surface area when the animal landed. For this purpose, we picked up the animal by the tail and lifted it vertically to a height of about 50 cm (lift duration 2 s; position holding at upper position: 1 s) and dropped it to a height of about 10 cm. When the animal touched the ground, we took a capture of the location of the four paws. Twenty repeated measurements were taken for each rat tested at each time point (pre-lesion, D1, 2, 3, 7, 10, 14, 21, and 30 post-lesion), and an average was calculated for each experimental session. The support surface was measured using a custom written image analysis tool (Matlab^®^, Mathworks, Incs, Natick, MS, USA). Data recorded after UVN were compared to the mean pre-lesion values using individual references, which means each animal acted as its own control.

### 2.9. Tissue Preparation

The rats were deeply anesthetized with a mixture of ketamine 1000/medetomidine and then perfused by intracardiac injection. The intracardiac injection of 400 mL of isotonic saline (0.9% NaCl) was followed by 400 mL of 4% paraformaldehyde in 0.1 M phosphate buffer (PB), pH 7.4. At the end of the perfusion, the brain was extracted from the skull and post fixed overnight at 4 °C in the same fixative solution as that used during the perfusion. Brains were rinsed and cryoprotected by successive transfers into increasing concentrations (10, 20, 30% of sucrose solution in 0.1 M PB for 72 h at 4 °C). Brains were rapidly frozen with CO_2_ gas and cut into serial 40 µm frontal section with a cryostat (Leica, Wetzlar, Germany) for immunochemistry.

### 2.10. Immunohistochemistry

Immunochemical labeling was performed according to previously validated protocols [7,8,10]. Cell proliferation was analyzed in groups 1 (UVN-NaCl-D3) and 3 (UVN-T4-D3) after injection of BrdU on D3 post-UVN, and the animals were killed 3 h later. Survival and differentiation of the newly generated cells were analyzed in groups 2 (UVN-NaCl) and 4 (UVN-T4) that were injected with BrdU on D3 post-UVN and killed on D30. Floating brain sections were washed (3 × 10 min) with PBS in multi-well plates. Blocking was done by incubation (1 h) in 5% BSA and 0.3% Triton X-100. Slides were incubated overnight at 4 °C with the following primary antibodies: mouse anti-BrdU (1:100, Dako, Santa Clara, CA, USA), rat anti-BrdU (1:100 Bio-rad, Marnes-la-Coquette, France), mouse anti-NeuN (1:100, Millipore, Burlington, MA, USA), rabbit anti-GFAP (1:200, Dako), rabbit anti-IBA1 (1:2000, Wako, Saitama, Japan), rabbit anti-Olig2 (1:500, Millipore), goat anti-Sox2 (1:100, R&D Systems Minneapolis, MN, USA), rabbit anti-TRα (1:200, Rockland), rabbit anti-TRβ (1:200, Invitrogen, Waltham, MA, USA), goat anti-DIO2 (1:200, Invitrogen), and rabbit anti-KCC2 (1:200, Millipore). Fluorescent secondary antibodies (Abcam, Cambridge, UK) were used as follows: Alexa Fluor 594 nm goat anti-rat (1:500), Alexa Fluor 594 nm donkey anti-goat (1:500), Alexa Fluor 488 nm donkey anti-rabbit (1:500), and Alexa Fluor 488 nm donkey anti-mouse (1:500) for 2 h at room temperature. Finally, brain sections were mounted onto SuperFrost/Plus glass slides (Fischer, Innviertel, Austria) and air-dried before being mounted with Roti^®^Mount FluorCare antifade reagent with the nuclear marker DAPI (Carl Roth, Karlsruhe, Germany).

### 2.11. Cells Count and Statistical Analysis

For quantification of cells expressing specific markers, 1 in 12 serial sections starting at the beginning of the vestibular nuclei (relative to bregma, −9.84 mm) to the end of the vestibular nuclei (relative to bregma, −13.08 mm), according to the rat brain stereotaxic atlas [30], were used. Only sections of the medial vestibular nucleus (MVN) on the lesioned (left) side were evaluated. Quantification of BrdU^+^ NeuN^+^, GFAP^+^, IBA1^+^, Olig2^+^, Sox2^+^, thyroid hormone receptor alpha^+^ (TRα), and thyroid hormone receptor beta^+^ (TRβ) cells was counted using confocal imaging with a Zeiss LM 710 NLO laser scanning microscope equipped with a 63×/1.32 NA oil immersion lens. For each marker, immunoreactive positive cells in the MVN were counted using an integrated microscopic counting chamber that delineated the region of interest by a square of 425.10 μm^2^. The average cell counts from the sections were used for statistical analysis.

### 2.12. Quantification of KCC2

A custom program written in Matlab^®^ (Mathworks, Incs) was developed to analyze the membrane fluorescence of KCC2 [31]. The background or non-specific immunofluorescence was evaluated by calculating the mean florescence in a selected area devoid of neurons or other immunolabeled structures. In this region, we then established a threshold equal to the mean immunofluorescence plus three times the standard deviation. All data were subtracted from this threshold value and only positive values were conserved for further analysis. A region of interest was then drawn around the plasma membrane of a neuron. The program calculated the mean membrane fluorescence of each cell body in the region of interest over data that were above 20% of the maximum value. This threshold was used to ensure that only pixels belonging to the plasmatic membrane were counted. The mean membrane fluorescence of the measured neurons enables us to estimate the level of membrane expression of KCC2. A low membrane expression of KCC2 is correlated with an increase in intracellular Cl^−^ ion concentration. The consequence of this increase is a decrease in the efficacy of GABAergic inhibition and, in the most severe cases, an exit of Cl^−^ ions from the cell by the action of GABA on its GABA_A_ receptor, thus converting GABAergic neurotransmission from inhibitory to excitatory [32].

### 2.13. Cytochrome Oxidase Histochemistry

Metabolic activity in the lateral vestibular nuclei was assessed by means of cytochrome oxidase (CO) labeling on series of 40 μm-thick coronal frozen sections. Slides were incubated overnight in the dark in a solution of 0.1 M phosphate buffer (pH 7.4) containing 10 mg of DAB (Tablet #D5905 Sigma-Aldrich, St. Quentin Fallavier, France) and 30 mg horse-heart cytochrome c (#C2506, Sigma-Aldrich, France) per 50 mL, stirred continuously (Sigma–Aldrich, St. Quentin Fallavier, France for all the chemical products). The slides were then washed two times in buffer (2 × 5 min), dehydrated in successive ethanol and xylene baths, and coverslipped with DPX. Images of the lateral vestibular nuclei, identified by the rat brain stereotaxic atlas [30], were collected and we counted the number of neurons positive to CO labeling only in the lateral vestibular nuclei.

### 2.14. Statistical Analysis

We performed all statistical analyses using GraphPad Prism software (version 7, GraphPad Software, San Diego, CA, USA). Summary graphs are all shown as mean ± SEM. A two-way repeated measures ANOVA with a post-hoc Bonferroni’s multiple comparison test were used to determine statistical differences between the treatments (UVN-NaCl and UVN-T4) for behavioral data. The statistical analyses of the cellular data were evaluated by two-way ANOVA to determine the effects of the group (UVN-NaCl and UVN-T4) and the post-operative time (Pre-op, D3 or D30) on BrdU^+^, NeuN^+^, GFAP^+^, IBA1^+^, Olig2^+^, Sox2^+^, TRα^+^, TRβ^+^, and CO^+^ cells to determine whether there were any interactions between these variables. ANOVA was followed by post hoc analysis with the Tukey’s test. *p* values of < 0.05 were considered as statistically significant. * *p* < 0.05, ** *p* < 0.01, *** *p* < 0.001.

## 3. Results

### 3.1. L-Thyroxine Treatment Reduced Postural Vestibular Deficits

After UVN, animals showed severe signs of posturo-locomotor alterations characteristic of vestibular syndrome that can be scored using different tests. L-T4 (10 µg/kg, i.p.) or 0.9% NaCl (vehicle, in equivalent volume, i.p.) was administered to rats at the end of the UVN procedure and a single injection per day over the first 3 days, 30 min before behavioral tests (Figure 1a).

These behavioral deficits were assessed at several time points over a period of 30 days using a qualitative ‘vestibular score’ scale (see Methods section). In both the UVN-NaCl and UVN-T4 groups, the mean vestibular score peaked on D1 (UVN-NaCl: 17 ± 0.53; UVN-T4: 12 ± 0.53) and never completely disappeared at up to 30 days after the lesion (Figure 1b). From D1 until D7, UVN-T4 treated animals had a lower vestibular score than UVN-NaCl animals (*p* < 0.0001).

Unilateral vestibular lesion also results in an increase in the support surface delimited by the animal’s four paws. This parameter is a good estimate of postural stability and restoration of balance [4,8,33,34]. When we compared the support surface area of each group between the pre-operative values and the values obtained after UVN (D1-D30), we observed that the UVN-NaCl group had significantly increased their support surface area the first 3 days following vestibular lesion (D1 to D3: *p* < 0.001) and reached pre-operative control values from D7 until D30 (Figure 1c). In contrast, the UVN-T4 group significantly increased their support surface area only at D1 after vestibular lesion (*p* < 0.001) restoring to pre-operative control values from D2. Comparison between groups showed that support surface area in the UVN-T4 group was reduced on D2 and D3 when compared to a saline injection after UVN (*p* < 0.01).

To further study postural impairment after UVN and the improvement by L-T4 treatment, we next tested the weight distribution of rats on the lateral axis. As shown in previous reports [28,33], before UVN, rats distributed their weight symmetrically between the right and the left sides (Figure 1d). After UVN, UVN-NaCl rats increased the weight applied on their ipsilesional paws, resulting in a postural asymmetry that reached 8% at D7 and increased to 11% at D30 (D7: *p* < 0.05; D30: *p* < 0.001; Figure 1d). In the UVN-T4 group, the postural asymmetry induced by UVN was not present (D7 and D30: *p* = ns). Taken together, these results suggest that L-T4 treatment improves the restoration of postural deficits induced by UVN.

### 3.2. L-Thyroxine Treatment Improved Locomotor Recovery after UVN

To investigate other behavioral alterations induced by L-T4 treatment after UVN, we conducted locomotor behavioral test batteries. We previously reported that UVN resulted in significant changes in locomotor activity [29]. In the open field test, UVN-T4 rats significantly reduced their total distance travelled only on D1 (*p* < 0.05) and did not present a hyperactivity phase from D10 to D30, as was observed in the UVN-NaCl group (Figure 2a,b). In contrast, the total distance travelled was remarkably diminished the first 3 days after UVN for the UVN-NaCl group (D1: *p* < 0.001; D2: *p* < 0.01; D3: *p* < 0.05) and conversely, increased from D10 until D30 (D10 to D30: *p* < 0.001). Similarly, the immobility time of UVN-NaCl rats increased the first 3 days after the lesion, with a maximum on D1 (D1 to D2: *p* < 0.001; D3: *p* < 0.01) before returning to pre-operative control values from D7 (Figure 2c). Immobility time in the UVN-T4 group increased only on D1 (*p* < 0.001), returned to control values by D2, and was reduced on D1 and D2 when compared to UVN-NaCl group (D1: *p* < 0.001; D2: *p* < 0.05). Mean velocity of the UVN-NaCl group (Figure 2d and Appendix A) was reduced the first 3 days after UVN (D1 and D2: *p* < 0.001; D3: *p* < 0.01) and increased from D10 to D30 (D10 to D30: *p* < 0.001). UVN-T4 rats did not present a reduction of velocity the first 3 days after UVN (Figure 2d and Appendix A) and were significantly different from UVN-NaCl rats on D1 and D2 (*p* < 0.01). However, the UVN-T4 group also adopted the locomotor strategy of increasing their velocity from D10 to D30 (D10 to D21: *p* < 0.05; D30: *p* < 0.01). In the same way, the number of high accelerations (Figure 2e) of the UVN-NaCl group was almost absent the first 3 days (D1 to D3: *p* < 0.001) and increased from D14 to D30 (D14: *p* < 0.01; D21 and D30: *p* < 0.05). In contrast, the UVN-T4 group was still able to achieve high accelerations during the first 3 days post-UVN, although with reduced accelerations on D1 and D2 when compared to pre-operative values (D1: *p* < 0.001; D2: *p* < 0.05). Furthermore, UVN-T4 treated rats more quickly increased their high accelerations (D7 vs. D14) when compared to the UVN-NaCl group (D7: *p* < 0.01). The number of high accelerations between groups was significantly different on D3 and D7 post-UVN (D3: *p* < 0.05; D7: *p* < 0.01). Finally, vestibular disorders induce imbalance and stumbling, which leads to difficulty in walking straight. The ‘meander’ parameter from Ethovision XT14 allowed us to quantify the difficulty of walking (Figure 2f). While animals injected with a saline solution had a drastically unstable gait on D1 post-UVN (D1: *p* < 0.001), animals injected with L-T4 did not exhibit significant gait instability. Collectively, these findings suggest that short-term L-T4 treatment reduces the vestibular syndrome after a unilateral vestibular loss and promotes functional recovery (Appendix A).

### 3.3. Presence of Thyroid Hormone Receptors and Thyroxine Deiodinase 2 in the Vestibular Nuclei

The results obtained from behavioral analysis after UVN show that L-T4 treatment improved the recovery of vestibular functions, and suggest an effect of L-T4 in the vestibular nuclei (VN). The genomic effects of TH are mediated by the interaction of T3, the active form of TH, with TH receptors acting as transcriptional factors regulating the expression of several genes [21,35]. A first approach to understand the mechanism of action of TH in the VN consisted of analyzing the expression of TRα, TRβ, and iodothyronine deiodinase type 2 (DIO2), which is responsible for converting T4 into T3 [36,37,38,39]. We found that VN of adult rodent brain expressed both TRα and TRβ, as well as the enzyme DIO2 (Figure 3a). We also observed SOX2-positive(^+^) and TR-negative(^−^) cells and vice versa (SOX2^−^/TRα^+^ cells, indicated by arrows; Figure 3b), which is strikingly similar to the pattern of neural stem cells reported in the subventricular niche [40]. This pattern confirms our previous results [10], revealing for the first time the presence of local neural stem cells in the vestibular nuclei of control rats. Taken together, these observations suggest that vestibular nuclei cells are able to provide their own intracellular T3 and display suitable molecular equipment to engage a genomic effect of thyroid hormones. Furthermore, the expression of both SOX2 and TRα in the vestibular nuclei may suggest an action of L-T4 on neural stem cell commitment and differentiation [40].

### 3.4. L-Thyroxine Up-Regulates TRα and Block the Up-Regulation of SOX2 in the Deafferented Vestibular Nuclei

Interestingly, TRα^+^ cells increased significantly at D3 after UVN only in animals treated with L-T4 when compared to basal levels or to the D3 UVN-NaCl group (Pre-op vs. D3 T4: *p* < 0.001; D3 NaCl vs. D3 T4: *p* < 0.01; Figure 4a,b). However, the up-regulation of TRα was present in both groups at D30 after UVN (UVN-NaCl: *p* < 0.01; UVN-T4: *p* < 0.05). Conversely, in UVN-T4 group, the number of SOX2^+^ cells at D3 post-UVN remained at the basal levels, while it increased in the UVN-NaCl group (*p* < 0.05; Figure 4a,c). As described in the subventricular neurogenic niche [40], the decreased number of SOX2^+^ and the increased number of TRα^+^ cells in the vestibular nuclei suggests that L-T4 injections may increase cell proliferation and neural differentiation by repressing the action of *Sox2* on activated vestibular neural stem cells [10]. To confirm this hypothesis, we evaluated cell proliferation with a single injection of BrdU (200 mg/kg) 3 h prior to intracardiac perfusion at D3 post-UVN and evaluated the cellular fate of the newly generated cells at D30 (see study design in Figure 5a).

### 3.5. L-Thyroxine Increases Cell Proliferation and Alters the Cellular Fate of Newly Generated Cells in the Deafferented Vestibular Nuclei

Unilateral vestibular loss induces numerous plastic events in the vestibular nuclei [3]. Among these events are strong cell proliferation, glial reaction, and reactive neurogenesis after UVN, which are all involved in vestibular compensation [5,6,7,8,41]. We recently demonstrated that quiescent neural stem cells in the MVN are reactivated after UVN [10] and that, interestingly, TH plays an important role in the proliferation and differentiation of neural stem cells [40,42]. Since TH affects glial cells’ function and governs many aspect of neurogenesis [11,12,39,43], we investigated whether L-T4 injections modulate reactive neurogenesis and glial reaction in the vestibular nuclei after UVN.

We evaluated cell proliferation with a single injection of BrdU (200 mg/kg) 3 h prior to intracardiac perfusion at D3 post-UVN (Figure 5a). The UVN-T4 group increased the mean number of BrdU^+^ cells in the deafferented MVN at D3 when compared to the UVN-NaCl group (*p* < 0.001; Figure 5b,c). However, the number of proliferative cells (BrdU^+^) in the UVN-T4 group at D30 was similar to the UVN-NaCl group.

We then evaluated the cellular fate of the newly generated cells (BrdU^+^) with a single injection of BrdU (200 mg/kg) at D3 post-UVN, prior to intracardiac perfusion at D30 (Figure 5a). Cellular markers were revealed by double immunohistochemical labeling presented in Figure 5d. In the MVN of the UVN-NaCl group, it was possible to determine the presence of newly generated neurons (BrdU^+^/NeuN^+^), astrocytes (BrdU^+^/GFAP^+^), and microglia (BrdU^+^/IBA1^+^) in almost equivalent proportion (~20%) with a predominance of the oligodendrocyte (BrdU^+^/Olig2^+^) phenotype (32%) (Figure 5e). Based on the literature, TH promotes the engagement of neural stem cells preferentially towards a neural fate [14,40,42,44]. Surprisingly, with our paradigm, we observed an increased formation of new microglial cells (41 vs. 24%) at the expense of new neurons (5 vs. 23%) in UVN-T4 group when compared to UVN-NaCl, respectively (Figure 5f).

### 3.6. L-Thyroxine Modulates the Glial Reaction and Prevents Downregulation of KCC2 in the Deafferented Vestibular Nuclei

As reported in our previous studies, UVN induces a significant increase in microglia (IBA1), astrocytes (GFAP), and oligodendrocytes (Olig2) at D3 post-lesion in the deafferented vestibular nuclei [5,6,7,8,10] (Figure 6a–d). The number of microglial cells was reduced in the UVN-T4 group when compared to UVN-NaCl rats at D3 post-lesion (*p* < 0.001; Figure 6a,b). However, the number of microglial cells in the UVN-T4 group was still significantly increased when compared to pre-operative values (*p* < 0.001). No difference in the number of astrocytes was found between the UVN-NaCl and UVN-T4 groups (Figure 6a,c). Finally, the number of oligodendrocytes at D3 was reduced in the UVN-T4 group when compared to both the UVN-NaCl group (*p* < 0.001) and pre-operative values (*p* < 0.01; Figure 6a,d). At D30 post-lesion, the number of oligodendrocytes for the UVN-T4 group was comparable with the UVN-NaCl group (Figure 6a).

We previously hypothesized that UVN-induced microglia upregulation triggered a microglia–BDNF–TrkB signaling pathway that decreased KCC2 expression in the deafferented vestibular nuclei [8]. This hypothesis of a downregulation of KCC2 by BDNF and microglia is supported by other studies [45,46,47]. As we showed a decrease in microglia with L-T4 injections at D3, we wondered if KCC2 expression was also altered. While UVN-NaCl animals had reduced KCC2 expression at D3 post-lesion when compared to pre-operative values (*p* < 0.001), no difference was found for the UVN-T4 group (Figure 6e,f). In addition, KCC2 expression in UVN-T4 rats at D3 post-UVN was increased when compared to UVN-NaCl rats on the same day (*p* < 0.001). Restoration of KCC2 expression was achieved one month after the lesion for the UVN-NaCl group. However, we observed a significant increase in KCC2 expression for UVN-NaCl rats when compared to UVN-T4 rats 30 days after UVN (*p* < 0.001; Figure 6f).

### 3.7. L-T4-Treated Rats Exhibit Enhanced Metabolic Activity in the Vestibular Nuclei Three Days after UVN

It has been known for a century that TH regulates energy metabolism [21,22,48]. Quantification of cytochrome oxidase (CO)-labeled neurons can provide insight into the metabolic effect of L-T4 in the vestibular nuclei after UVN. We focused on the lateral vestibular nucleus (LVN) in view of the extensive vestibulo-spinal projections of this nucleus implicated in gait and balance [49,50], both at an acute (D3) and a compensated stage (D30) after UVN (summary of the study design presented in Figure 7a). The number of CO-labeled neurons in UVN-T4 group on both ipsi- and contralesional LVN is increased when compared to both sides of the LVN in the UVN-NaCl group at D3 post-lesion (*p* < 0.001; Figure 7b,c). One month after the lesion, both groups had a lower number of CO-labeled neurons, but the number of CO-labeled neurons on the ipsilesional LVN in the UVN-T4 group had increased when compared to the same side of LVN in the UVN-NaCl group (*p* < 0.05; Figure 7b,d). These data suggest that L-T4 injections enhanced metabolic activity in the LVN during the first days after the lesion, which is consistent with the behavioral results.

## 4. Discussion

The effect of a short-term L-T4 treatment on the vestibular syndrome following UVN struck: all the vestibular behavioral parameters we analyzed, showing a significant improvement. Our qualitative assessment of the vestibular syndrome indicated about 30% reduction in symptoms in the UVN-T4 group at the first day after the lesion. In addition, L-T4-treated animals reached the plateau of vestibular compensation earlier than untreated animals (between 3 and 7 days vs. 10 days). Based on our previous work [28,29,33], we analyzed more specifically the posturo-locomotor syndrome and demonstrated that L-T4 treatment promoted a faster recovery of postural and locomotor functions that had been impaired by UVN. The reported improvements in walking speed and acceleration in L-T4-treated rats can be directly transposed to the clinic. Increasing walking speed for vestibular patients is considered as an improvement during a vestibular rehabilitation program [51,52,53]. Furthermore, changes in gait speed and accelerations can be used to assess dynamic postural stability (e.g., by Dynamic Gait Index) [54], and can be used in vestibulospinal exercises for vestibular rehabilitation [53,55,56]. The future clinical implications of the beneficial effect of TH on vestibular compensation for therapy of patients with acute vestibular loss can be two-fold: (1) Similarly to the current rat experiments, L-T4 could be administered in the acute phase of vestibular syndrome to reduce symptoms of vestibular asymmetry and accelerate vestibular compensation. The translation to clinical practice is facilitated by the widespread availability of L-T4 medication; (2) pre-existing hypothyroidism may be considered as a risk factor for poor vestibular compensation or vertigo, which should be screened more regularly [57,58,59,60].

The TH is represented by T3 and T4 molecules; moreover, the principal product of the thyroid gland is T4, which is converted into T3 by the DIO2 enzyme. The genomic effects of TH are mediated by the ability of TH nuclear receptors (TRα and TRβ) to interact directly with gene promoters in the presence or absence of their ligand, which is primarily T3. In the present study, we observed short (from D1 to D3) but also long-term effects (at D30). Although a short-term effect is unlikely for a hormone that acts by regulating gene expression, the obtained results support a more rapid mode of action of TH. Over the past years, many different studies have illustrated the possibility of a rapid response and several alternative, so-called non-genomic pathways for TH action (for review see: [21,61]). Indeed, TH could participate in signaling pathways independently of direct or indirect DNA binding or can act independently of the TH receptor [35] through the integrin αVβ3 receptor [62]. Despite the fascinating physiological effect of L-T4 on vestibular function recovery in UVN rats, the intimate mechanisms by which thyroid hormones act remain to be elucidated, but we can hypothesize the different pathways involved.

The different symptoms observed after unilateral vestibular loss result from asymmetry of spontaneous electrical activity between homologous VN [63,64]. The data in the literature show that the restoration of electrophysiological equilibrium between the two opposite VN is the key parameter for the functional recovery of postural, locomotor, and gaze stabilization functions [2,65,66]. TH is known to stimulate diverse metabolic activities, resulting in increased ATP and oxygen consumption [22,67,68]. In line with the literature, the L-T4-treated group exhibited a bilateral increase in the number of CO-labeled neurons in the LVN three days after UVN when compared to the UVN-NaCl group. Moreover, the increased number of CO-labeled neurons observed in the deafferented LVN in the UVN-T4 group was significantly higher than that observed in the contralesional LVN of the vehicle-treated group, highlighting a drastic increase in metabolic activity due to L-T4 treatment. In relation to the increase in metabolic activity of neurons, T3 and T4 administration reduced GABA-evoked current and modulate inhibitory neurotransmission by non-genomics mechanisms [69,70]. Thus, stimulation of metabolic activity and inhibition of GABAergic neurotransmission by L-T4 treatment could increase the excitability of ipsilesional vestibular neurons, thereby reducing the electrophysiological asymmetry between homologous vestibular nuclei, which attenuates the syndrome and accelerates vestibular compensation. Furthermore, the fact that we observed an increase in the expression of CO on the healthy side can be explained by an action of L-T4 on local inhibitory or commissural excitatory neurons (with projection towards the injured side). This action of L-T4 on the intact side would also participate in the rebalancing of activity between homologous VN. The architecture of the vestibular neural networks has already been well explained in a previous article [4].

Interestingly, the deafferented vestibular environment has been reconfigured to ensure the binding of endogenous and exogenous thyroid hormones by increasing TRα expression in the UVN-T4 group. As DIO2 enzyme is mainly expressed in astrocytes [37], we can hypothesize that the astroglial reaction observed in UVN animals actively participates in the conversion of T4 in T3 and supports all the mechanisms promoting vestibular compensation in response to the metabolic energy demand of the deafferented VN. Indeed, a proteomic analysis in the MVN of rats one week after unilateral labyrinthectomy (UL) showed an increase in proteins associated with mitochondrial function and energy metabolism [71]. One of these proteins was succinate dehydrogenase, a crucial enzyme in the mitochondrial Krebs cycle, for which positive regulation was observed at 5 and 15 days after UVN in the ipsilesional LVN of rabbits [72]. Overall, these changes reflect the high metabolic energy demand of the deafferented VN, which is associated with glial response, cell proliferation, neurogenesis, and structural remodeling to compensate for the loss of vestibular input.

Based on other studies [45,46,47], we have proposed that UVN-induced upregulation of microglia triggers a microglia–BDNF–TrkB signaling pathway that transiently decreases KCC2 expression in the deafferented VN of the feline model [8]. We postulated that this acute and transient KCC2 remodeling promotes a depolarizing action of GABA, facilitating the restoration of the excitability level in the deafferented vestibular environment that is crucial for vestibular function recovery. In the present study, we reproduced this result on the UVN rodent model, but we demonstrated that L-T4 treatment prevents the negative regulation of KCC2 in the deafferented LVN when compared to the UVN-NaCl group. A first explanatory hypothesis is that recourse to this transient KCC2 downregulation, with a view to restoring the excitability of deafferented VN, is no longer necessary. Indeed, the energy metabolism increased by T4 treatment can probably alone support the restoration of the level of excitability, underlying a faster functional restoration. However, an opposite action of BDNF on KCC2 expression has been described in axotomized corticospinal neurons [73]. Whereas in intact mature neurons BDNF downregulates KCC2 [47,74,75], in injured neurons, BDNF upregulates KCC2 [8,73,74]. Interestingly, TH have been shown to increase the expression of BDNF in lesioned slices of the hippocampus [18,76] and other pathological models [15,16,17]. We can then assume that a short-term L-T4 treatment after UVN induces a local increase of BDNF in the deafferented VN, thus blocking KCC2 downregulation and promoting cell proliferation as demonstrated in our previous study in the UVN feline model [8]. Finally, the absence of KCC2 downregulation induced after L-T4 treatment is consistent with the hypothesis of Kaila et al. [32], which proposes that “downregulation of KCC2 following neuronal trauma may be part of a general adaptive cellular response that facilitates neuronal survival by reducing the energetic costs that are needed to preserve low [Cl^−^]_i_”. By providing TH, known to increase energy metabolism, the internalization of KCC2 to reduce energy costs after UVN would no longer be necessary.

Unsurprisingly, L-T4 injections after UVN reduce the number of Olig2^+^ cells and prevent the increase of SOX2^+^ cells in the deafferented VN. These results are congruent with the literature [40,77], and are in agreement with the increased cell proliferation observed in the UVN-T4 group three days after the lesion. However, in our paradigm we did not observe cell differentiation preferentially towards a neuronal fate (BrdU^+^/NeuN^+^ cells) with the L-T4 treatment. On the contrary, cell differentiation was promoted towards a microglial cell phenotype (Brdu^+^/IBA1^+^ cells) in the UVN-T4 group. This result could appear paradoxical, since TH promotes the engagement of neural stem cells preferentially towards a neuronal fate [14,40,42,44]. Yet, our results do not contradict this well-established concept. As expected, L-T4 injections during the first three days post-UVN significantly increased cell proliferation on the third day without impacting the cell survival ratio. It is noteworthy that cell differentiation and survival in the present study were evaluated at day 30, which is 4 weeks after the last L-T4 injection. Given the T4 half-life of 12–24 h in rats [78], the treatment during the first 3 days after UVN had been eliminated from the system when cell differentiation and survival was assessed (at D30 after UVN). We believe that an extended treatment window would have revealed an increased cell survival and neural differentiation as described in the literature. Another explanation could come from the reduction of the vestibular syndrome observed with L-T4 treatment in the first days after injury. Neurogenesis is indeed expressed in the UVN model, but not in the labyrinthine lesion or hair cell destruction model [6]. What differentiates these three models is the severity of the vestibular syndrome; it is most severe in the UVN model and weaker in the other two models. Since the UVN–vestibular syndrome is considerably reduced by L-T4 treatment and the vestibular compensation is accelerated (more akin to the kinetics of a labyrinthine lesion model), it is possible that neuronal differentiation is not necessary to compensate for the deficits. In agreement with this hypothesis, we have demonstrated a decrease in the percentage of BrdU^+^/NeuN^+^ in cats treated with BDNF, showing an improvement in vestibular compensation [8]. With this hormonal treatment, we provide other compensation pathways to the central nervous system, notably based on energy metabolism and excitability, which could explain the limited number of newly generated neurons we observed in our model.

We demonstrated, in this study, that L-T4 treatment promotes microglial rather than neuronal differentiation. It is interesting to mention that the same result was recently observed in the same UVN rodent model subjected to a sensorimotor rehabilitation protocol [79]. Likewise, T3 increases the number of microglia cell bodies, promotes microglia survival, and enhances the growth of their processes [80]. Like neurogenesis, microglia also appears as a factor interacting with different plasticity mechanisms of the deafferented vestibular environment (excitability, inflammation, neuroprotection, etc.) promoting vestibular function recovery [79].

Obviously, the results obtained support a local action of TH in the deafferented VN. However, given the intraperitoneal administration, we cannot exclude an effect on the whole central network of vestibular compensation [81,82] as well as peripheral effects. Skeletal muscle is in fact a primary target for TH signaling in human and rodents [83]. TH are involved in contractile function by increasing the rate of muscle relaxation–contraction, in energy metabolism by increasing mitochondrial biogenesis, but also in myogenesis and muscle regeneration [83,84]. Interestingly, UVN induces asymmetry of muscle tone with ipsilesional hypotonia and contralesional hypertonia [24,85], thus altering the postural and locomotor balancing function. While no studies to date have demonstrated an increase in muscle tone after TH administration, it is plausible to hypothesize that by increasing contractile function and muscle metabolism, L-T4 treatment could accelerate the rebalancing of muscle tone after UVN, which is crucial for postural stability.

In summary, short-term administration of L-T4 after unilateral vestibular loss greatly reduces vestibular syndrome and improves vestibular compensation. The present results demonstrate different mechanisms of action of L-T4 underlying the antivertigo effect and open new perspectives of L-T4 treatment for acute vestibular loss.

### Clinical Relevance

The UVN model consists of a complete surgical section of the vestibular nerve unilaterally [24,86]. It therefore first produces the neurophysiological conditions encountered in vestibular neurotomy [87,88]. It can be anticipated that the antivertigo effect revealed in the present study may benefit patients undergoing this type of intervention. Further pre-clinical studies in the UVN rodent model should help define the most appropriate dosage and administration window before proceeding to clinical trials.

The etiological mechanisms leading to peripheral vestibulopathies are not fully established so far. However, at least some of these conditions may result from partial deafferentation of peripheral vestibular sensors unilaterally. Although it is currently unknown to what extent these peripheral damage impacts the central vestibular networks. It could be interesting to test the effect of L-T4 on the different types of vestibular deafferentation models available in rodents [86,89,90,91], to evaluate its possible use in human pathologies such as acute peripheral vestibulopathy, or Ménière’s disease. Also, other pharmacological compounds mimicking or modulating the effect of L-T4 should be tested in order to identify potential drug candidates to relieve peripheral vestibulopathies.

## Figures and Tables

**Figure 1 cells-11-00684-f001:**
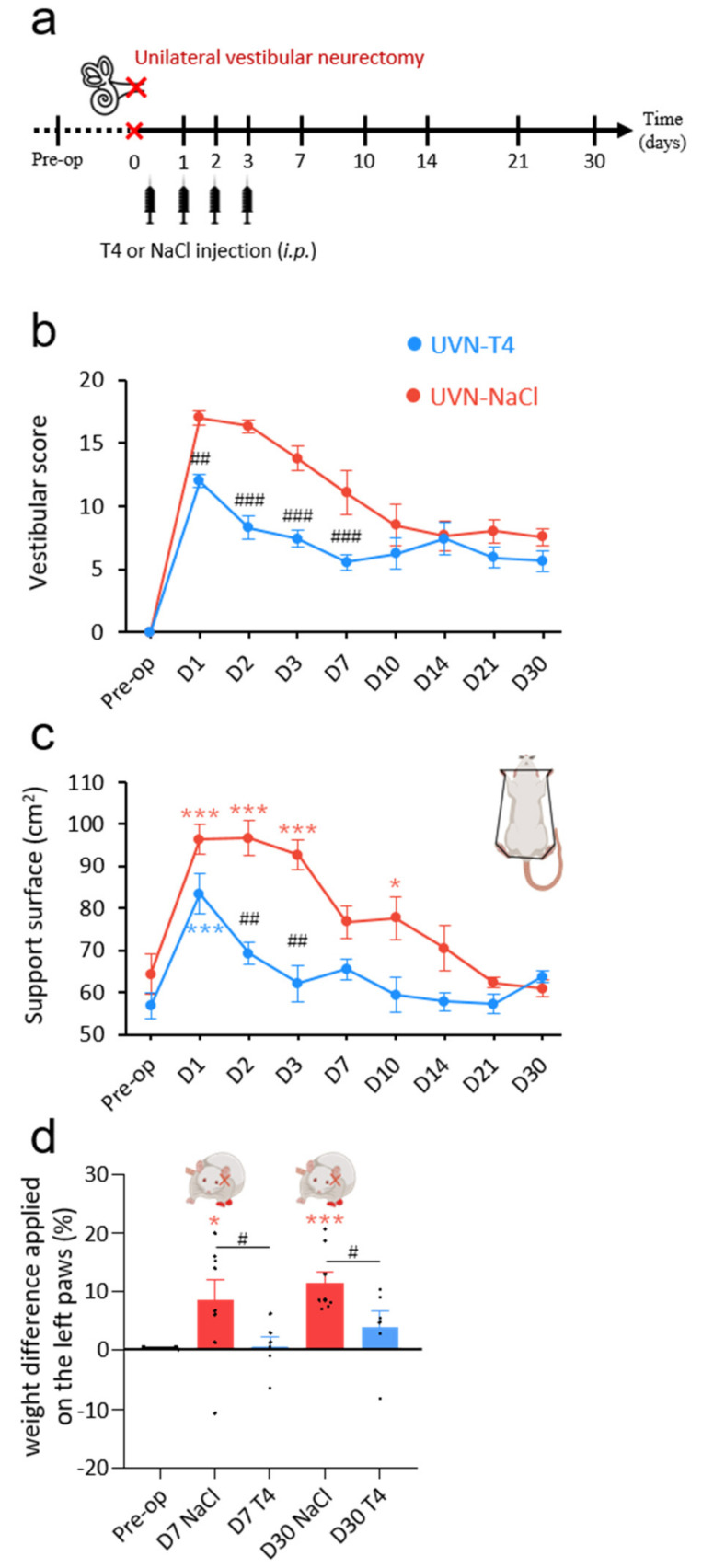
L-T4 treatment accelerates recovery from UVN-induced postural alterations. (**a**) Study design used to assess the behavioral effects of acute L-thyroxine (L-T4) treatment after unilateral vestibular neurectomy (UVN). UVN-T4 group received L-T4 injections (10 µg/kg dissolved in saline, i.p.) and UVN-NaCl group received 0.9% NaCl (vehicle, in equivalent volume, i.p.) at the end of the UVN and one injection/day during the first 3 days post-lesion. (**b**) Curves illustrating the time course (on the abscissae) of the vestibular syndrome (vestibular score on the ordinates) of UVN-NaCl (red) and UVN-T4 (blue) groups. (**c**) Curves indicating the mean post-operative recovery of the support surface of UVN-NaCl and UVN-T4 groups. (**d**) Histograms illustrating the weight distribution on the lateral axis of UVN-NaCl and UVN-T4 groups. Error bars represent SEM. A significant difference from the pre-operative value is indicated by * in red for UVN-NaCl group. A significant difference from the pre-operative value is indicated by * in blue for UVN-T4 group. A significant difference between UVN-NaCl (*n* = 8) and UVN-T4 (*n* = 7) group is indicated with # in black (* *p* < 0.05, ** *p* < 0.01, *** *p* < 0.001; repeated measure two-way ANOVA and Bonferroni’s *post hoc* tests).

**Figure 2 cells-11-00684-f002:**
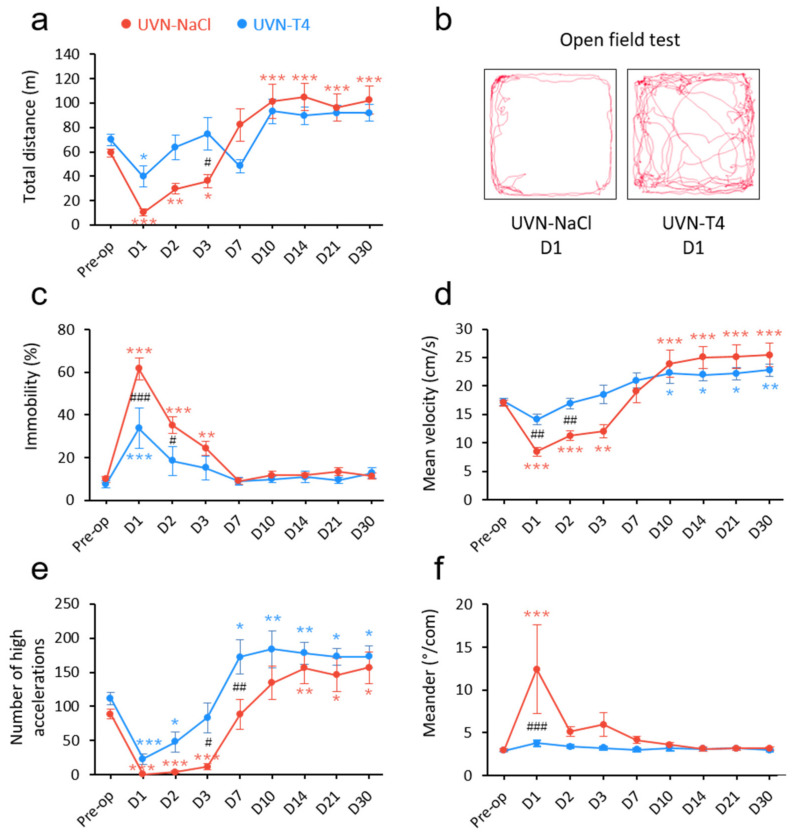
L-T4 treatment accelerates recovery from UVN-induced locomotor alterations. (**a**) Curve illustrating the mean post-operative recovery of the total distance moved by UVN-NaCl and UVN-T4 rats in the open field. (**b**) Video-tracked path of a UVN-NaCl and a UVN-T4 rat one day after UVN in the open field over a 10 min period. (**c**) Curves illustrating the kinetics of the percentage of time when the animal is immobile of UVN-NaCl (red) and UVN-T4 (blue) groups. (**d**) Curves illustrating the kinetics of the mean velocity (cm/s) of the animals (calculated from the center-point) of UVN-NaCl and UVN-T4 groups. (**e**) Curves illustrating the kinetics of the mean number of high accelerations (above 50 cm/s^2^) of the animals (calculated from the center point) of UVN-NaCl and UVN-T4 groups. (**f**) Curves illustrating the mean post-operative recovery of meander (sinuous trajectory) in the UVN-NaCl and UVN-T4 groups. Error bars represent SEM. A significant difference from the pre-operative value is indicated by * in red for UVN-NaCl group. A significant difference from the pre-operative value is indicated by * in blue for UVN-T4 group. A significant difference between UVN-NaCl (*n* = 8) and UVN-T4 (*n* = 7) group is indicated with # in black (* *p* < 0.05, ** *p* < 0.01, *** *p* < 0.001; repeated measure two-way ANOVA and Bonferroni’s *post hoc* tests).

**Figure 3 cells-11-00684-f003:**
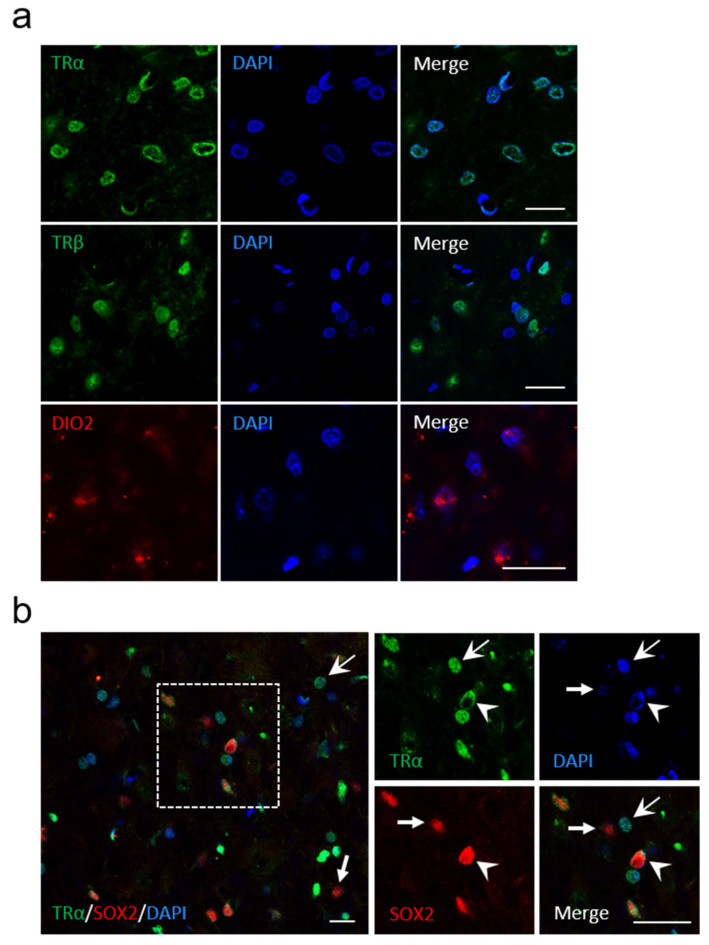
Presence of TRα, TRβ and Dio2 in the vestibular nuclei of control rats. (**a**) confocal immunostaining images confirming the presence of TRα (green), TRβ (green), and Dio2 (red) in the vestibular nuclei of control animals (without UVN). (**b**) Colocalization of TRα (green), SOX2 (red), and DAPI (blue) if cells in the vestibular nuclei. Head arrows indicate TRαPL/SOX2^+^/DAPI^+^ cells, arrows with tail indicate TRα^+^/SOX2^−^/DAPI^+^ cells, and small arrows with a tail indicate TRα^−^/SOX2^+^/DAPI^+^ cells. Scale bar = 20 µm.

**Figure 4 cells-11-00684-f004:**
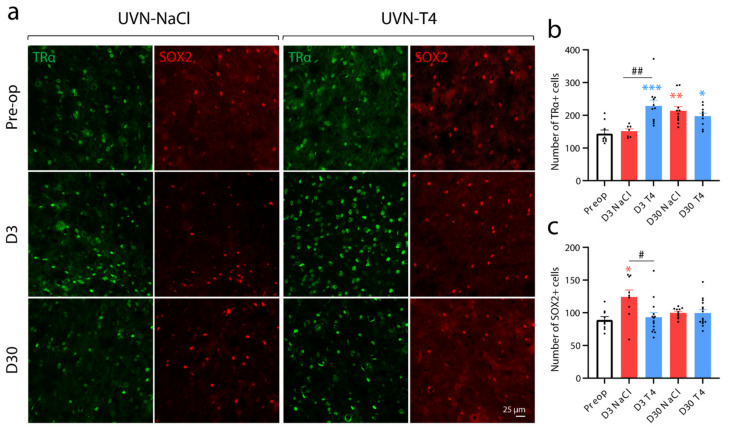
L-T4 treatment prevents the positive regulation of SOX2^+^ cells and increased the number of TRα^+^ cells three days after UVN in the deafferented medial vestibular nucleus. (**a**) Confocal immunostaining images of TRα^+^ (green) and SOX2^+^ (red) cells in the deafferented medial vestibular nucleus (MVN) before the lesion (pre-op), three days (D3) or thirty days (D30) after UVN for UVN-NaCl and UVN-T4 groups. (**b**) Quantitative assessment of the effect of UVN and L-T4 treatment on the number of TRα^+^ cells in the deafferented MVN. (**c**) Quantitative assessment of the effect of UVN and L-T4 treatment on the number of SOX2^+^ cells in the deafferented MVN. Error bars represent SEM. A significant difference from the pre-operative value is indicated by * in red for UVN-NaCl group. A significant difference from the pre-operative value is indicated by * in blue for UVN-T4 group. A significant difference between UVN-NaCl and UVN-T4 group is indicated with # in black (* *p* < 0.05, ** *p* < 0.01, *** *p* < 0.001; one-way ANOVA and Tuckey’s *post hoc* tests). Scale bar = 25 µm.

**Figure 5 cells-11-00684-f005:**
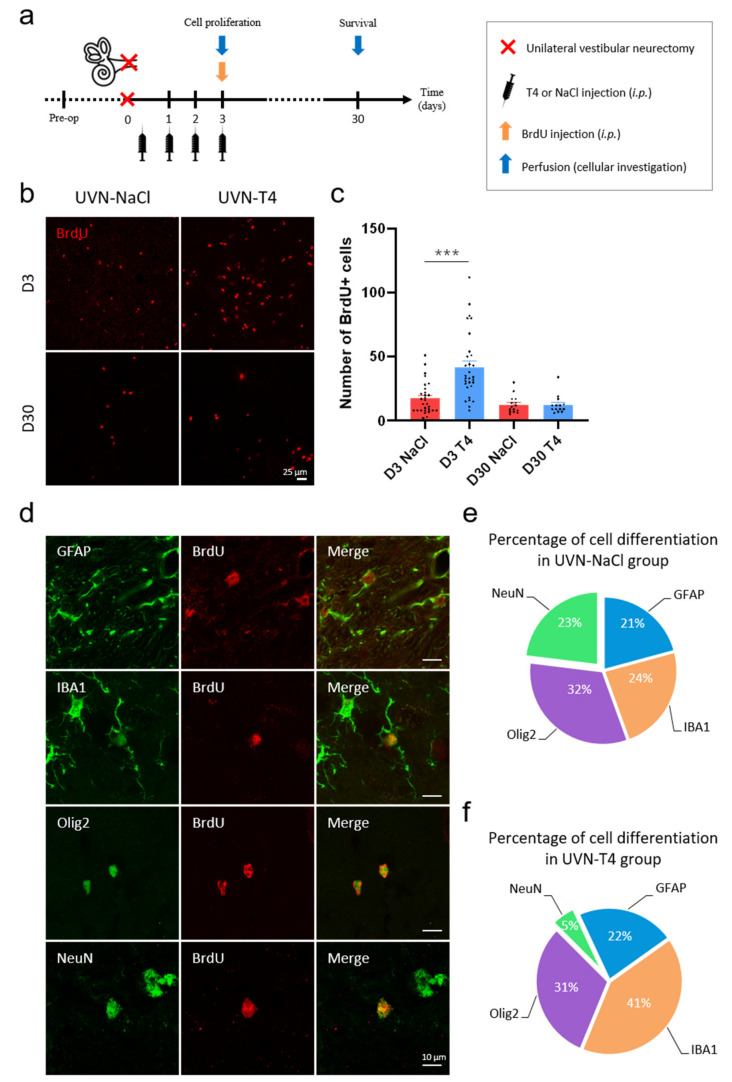
L-T4 treatment modulates cell proliferation and neurogenesis after UVN. (**a**) Study design used to assess cell proliferation and survival of the newly generated cells after unilateral vestibular neurectomy (UVN) and L-thyroxine (L-T4) treatment. UVN-T4 group received L-T4 injections (10 µg/kg, i.p.) and the UVN-NaCl group received 0.9% NaCl (vehicle, in equivalent volume, i.p.) at the end of the UVN and one injection/day during the first 3 days post-lesion. At day 3, animals received a BrdU injection (200 mg/kg, i.p.) and were perfused either 3 h later to evaluate cell proliferation, or on day 30, to investigate survival and differentiation of the newly generated cells. (**b**) Confocal immunostaining images of BrdU^+^ cells in the deafferented medial vestibular nucleus (MVN) before the lesion (pre-op), three days (D3) and thirty days (D30) after UVN for UVN-NaCl and UVN-T4 groups. Scale bar = 25 µm. (**c**) Quantitative assessment of the effect of UVN and L-T4 treatment on the number of BrdU^+^ cells in the deafferented MVN of UVN-NaCl (red, *n* = 4/delay) and UVN-T4 (blue, *n* = 4/delay) groups. (*** *p* < 0.001; one-way ANOVA and Tuckey’s *post hoc* tests). (**d**) Maximum intensity projection of Z-stack confocal images of cell differentiation evaluated in the deafferented MVN 30 days after UVN. The BrdU nuclei are in red, and the other markers of differentiation are: GFAP (astrocyte), IBA1 (microglia), Olig2 (oligodendrocytes), and NeuN (neuron) are in green. Scale bar = 10 µm. (**e**,**f**) Proportions of new neurons (green), astrocytes (blue), microglia (orange), and oligodendrocytes (purple) generated in the deafferented MVN of the UVN-NaCl (**e**) and UVN-T4 (**f**) groups 30 days after the lesion.

**Figure 6 cells-11-00684-f006:**
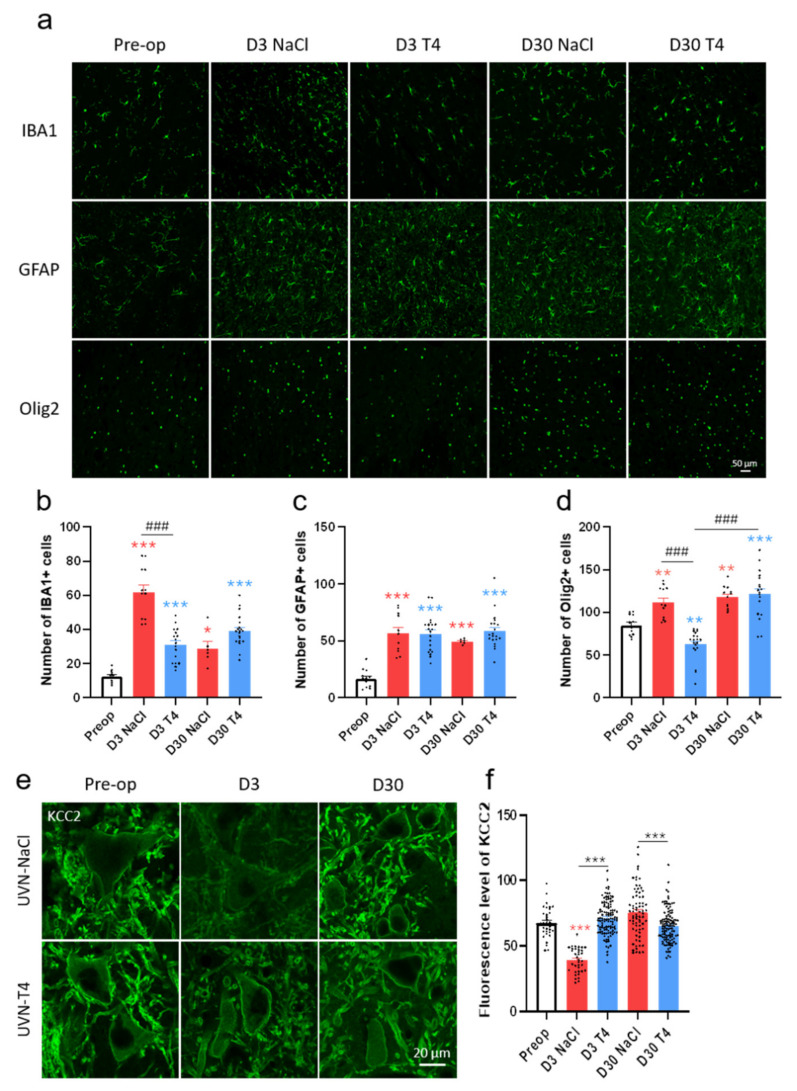
L-T4 treatment modulates glial reaction in the deafferented medial vestibular nucleus after UVN. (**a**) Confocal immunostaining images of IBA1^+^, GFAP^+^, and Olig2^+^ cells in the deafferented medial vestibular nucleus (MVN) before the lesion (pre-op), three days (D3), and thirty days (D30) after unilateral vestibular neurectomy (UVN) for UVN-NaCl and UVN-T4 groups. Scale bar = 50 µm. (**b**–**d**) Quantitative assessment of the effect of UVN and L-T4 treatment on the number of IBA1^+^ (**b**), GFAP^+^ (**c**), and Olig2^+^ (**d**) cells in the deafferented MVN of UVN-NaCl (red, *n* = 4/delay) and UVN-T4 (blue, *n* = 4/delay) groups. (**e**) Confocal immunostaining images in the deafferented lateral vestibular nucleus showing the KCC2 staining (green) in vestibular neurons of UVN-NaCl and UVN-T4 groups, observed prior to lesioning (pre-op, *n* = 4), D3 (*n* = 4) and D30 (*n* = 4) after UVN. (**f**) Quantification of the density of membrane labeling in vestibular neurons of intact rats and in UVN-NaCl (red) and UVN-T4 (blue) groups at either D3 or D30 after UVN. Scale bar = 20 µm. Error bars represent SEM. A significant difference from the pre-operative value is indicated by * in red for UVN-NaCl group. A significant difference from the pre-operative value is indicated by * in blue for UVN-T4 group. A significant difference between UVN-NaCl and UVN-T4 group is indicated with # in black (* *p* < 0.05, ** *p* < 0.01, *** *p* < 0.001; One-way ANOVA and Tuckey’s *post hoc* tests).

**Figure 7 cells-11-00684-f007:**
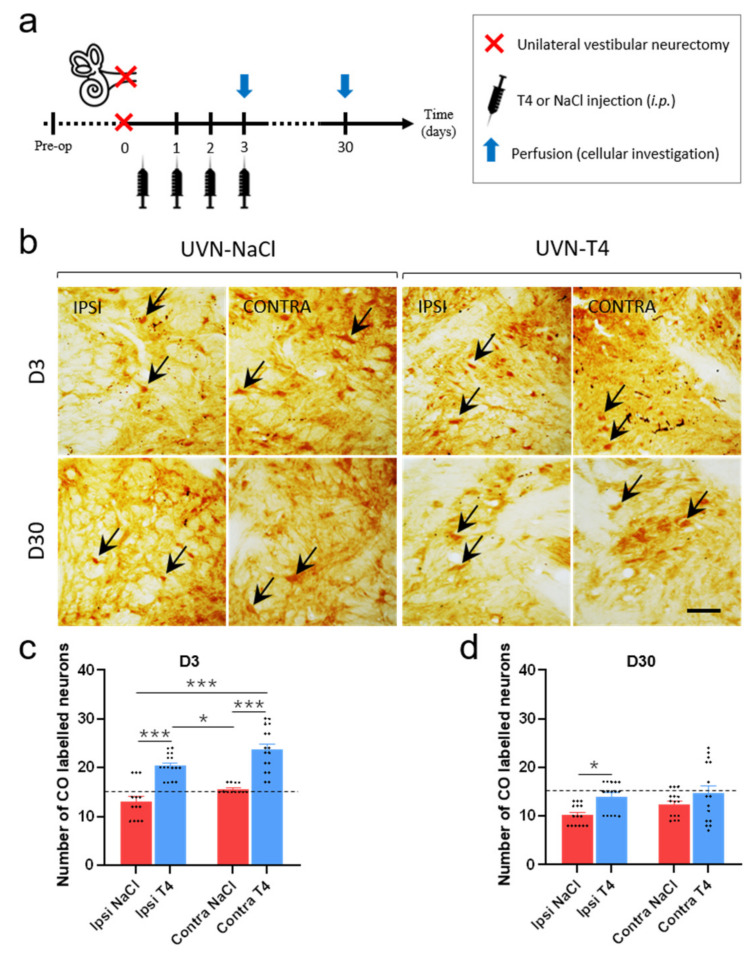
L-T4 treatment after UVN enhances metabolic activity in the vestibular nuclei. (**a**) Study design used to assess cellular consequences of acute L-thyroxine (L-T4) treatment after unilateral vestibular neurectomy (UVN). UVN-T4 group received L-T4 injections (10 µg/kg, i.p.) and UVN-NaCl group received 0.9% NaCl (vehicle, in equivalent volume, i.p.) at the end of the UVN and one injection/day during the first 3 days post-lesion. Animals were perfused either at an acute stage of vestibular compensation (3 h after the BrdU injection on day 3) or at a compensated stage (on day 30). (**b**) Illustrations of cytochrome oxidase (CO) enzymatic labeling in the lateral vestibular nucleus (LVN) of the UVN-NaCl (*n* = 4/delay) and UVN-T4 (*n* = 4/delay) groups observed at three days (D3) and thirty days (D30) after unilateral vestibular neurectomy (UVN). Illustrations are shown on both the contralesional and ipsilesional (deafferented) side. (**c**–**d**) Quantitative assessment of the effect of UVN and L-T4 treatment on the number of CO-labeled neurons in the LVN at D3 (**c**) or D30 (**d**). Dashed line represents the pre-operative value of the number of CO-labeled neurons in the LVN. Error bars represent SEM. A significant difference between UVN-NaCl and UVN-T4 group is indicated with * in black (* *p* < 0.05, *** *p* < 0.001; one-way ANOVA and Tuckey’s *post hoc* tests). Black arrows indicate example of CO^+^ neurons. Scale bar = 25 µm.

## Data Availability

The data presented in this study are available on request from the corresponding author.

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
