# Peer review of "L-Thyroxine Improves Vestibular Compensation in a Rat Model of Acute Peripheral Vestibulopathy: Cellular and Behavioral Aspects"

_cells, 2022, doi:10.3390/cells11040684_

Round 1
Reviewer 1 Report
The authors conduct a preliminary exploration about the effects of short-term L-T4 injections on vestibular compensation in a rat model of unilateral vestibular neurectomy by measuring the altered process of vestibular recovery directly through various qualitative behavioral assessments and indirectly through the level of neurogenesis. The role of T4 in vestibular plasticity proposed by this work offers some new insights into this field and may stimulate further work on the detailed underlying mechanisms. The hypothesis is interesting but there are still several problems that need to be improved in this work.
- In Figure 6, the authors compared the number of CO-labelled neurons between the UVN-T4 and UVN-NaCl groups to figure out the metabolic effects of L-T4 treatment on vestibular compensation. Since T4 itself could increase the overall metabolism rate which may be irrelevant with vestibular compensation procedure, the authors should better further analyze the difference among UVN-T4, UVN-NaCl, SHAM-T4, and SHAM-NaCl to correct the base alteration induced by T4 injection. Figuring out the metabolic difference between ipsi- and contra- lateral MVN within each group could also offer more information to illustrate the effect of T4 injection.
- As the authors found the most obvious effect during the first 3 days, it may be more important to show the difference of cell differentiation in D3 rather than D30 in figure 4. The result should be added and the D30 results could be further used for analysis and discussion on its long-term effects.
3. In Figure 5 a, e, and f, the number of Iba1+ cells are less in the T4 group while it shows a higher IF intensity of KCC in D3, which seems to conflict with the original hypothesis that the gliosis of microglia decreases the KCC expression and thus facilitate the vestibular compensation. It may be better to include more discussion.
- It could be better to offer detailed information on the vestibular recovery procedure measured by the weight difference applied on the left paws. The tendency of recovery is obscure here now, as the weight applied on the left paws tends to keep increasing.
- Similar to Question 1 mentioned above, T4 could increase the metabolic activity of the whole body. Is there a systematic effect on behavioral scores with the administration of T4 30 minutes before behavioral tests? (Especially in open field test) It may be better to partially exclude the influence by analyzing the difference among UVN-T4, UVN-NaCl, SHAM-T4, and SHAM-NaCl groups.
- It must be noted that Supplement Figure 1. A should be corrected since the IF picture of TR and SOX2 in UVN-NaCl and UVN-T4 group at pre-operation are exactly the same one. Although the base level of 2 groups is supposed to be the same, it’s not appropriate to use the same picture.
- More illustration may be needed in the result description regarding to the meaning and importance of the co-localization of Sox2 and TRa in Figure 3.
Author Response
Comments and Suggestions for Authors: Reviewer 1
The authors conduct a preliminary exploration about the effects of short-term L-T4 injections on vestibular compensation in a rat model of unilateral vestibular neurectomy by measuring the altered process of vestibular recovery directly through various qualitative behavioral assessments and indirectly through the level of neurogenesis. The role of T4 in vestibular plasticity proposed by this work offers some new insights into this field and may stimulate further work on the detailed underlying mechanisms. The hypothesis is interesting but there are still several problems that need to be improved in this work.
- In Figure 6, the authors compared the number of CO-labelled neurons between the UVN-T4 and UVN-NaCl groups to figure out the metabolic effects of L-T4 treatment on vestibular compensation. Since T4 itself could increase the overall metabolism rate which may be irrelevant with vestibular compensation procedure, the authors should better further analyze the difference among UVN-T4, UVN-NaCl, SHAM-T4, and SHAM-NaCl to correct the base alteration induced by T4 injection. Figuring out the metabolic difference between ipsi- and contra- lateral MVN within each group could also offer more information to illustrate the effect of T4 injection.
The decision to omit in present study a SHAM-NaCl group was based first upon ethical considerations (3R rule) with the aim to spare a whole group of 8 animals, second because this group has been extensively published in the same model (Marouane et al., 2020; Péricat et al., 2017; Rastoldo et al., 2020) and third because the present sham surgery does not induce any behavioral deficit evaluated with our behavioral tests: the animals remains stable across time in absence of neurectomy (Marouane et al., 2020).
In this study our main objective was to evaluate the pharmacological effects of a short-term thyroxine treatment on a rat model of vestibulopathy. Indeed, reduction of metabolic staining has been observed in the ipsilateral VN of unilateral labyrinthectomized rats (Llinas and Walton, 1979; Luyten et al., 1986) and cats (Maeda, 1988). Our data in the UVN cat point to similar cyotochome oxydase reductions in the deafferented lateral vestibular nucleus during the acute stage of the lesion and a return toward symmetry in staining was observed in the compensated animals (Lacour and Tighilet, 2010). Considered together, these findings suggest that vestibular loss leads to a reduction of energy demand for oxidative metabolism in the VN, very likely as a result of a concomitant reduction in neuronal activity.
Considering the functional role of thyroxine in energy metabolism, we opted for a marker of brain energy metabolism: the cytochrome oxidase. The different symptoms observed after unilateral vestibular loss result from asymmetry of spontaneous electrical activity between homologous vestibular nuclei (VN). The data in the literature show that the restoration of electrophysiological equilibrium between the two opposite VN is the key parameter for the functional recovery of postural, locomotor and gaze stabilization functions. We expected that L-T4 injections would increase metabolic activity on the injured side by targeting low activity neurons in order to restore the electrophysiological balance and promote vestibular compensation, which may spare us the use of a SHAM-T4 group and comparison to UVN-T4 group.
We focused our analysis on the lateral vestibular nucleus (LVN) in view of the extensive vestibulo-spinal projections of this nucleus implicated in gait and balance (McCall et al., 2017; Murray et al., 2018). The LVN contains both small cells and large neurons (giant cells of Deiters) which can be easily observed under the microscope as shown in Figure 6. In order to facilitate the counting of CO-positive cells and because of the vestibulo-spinal tract in the LVN we chose to focus only on this vestibular nucleus. However, we suppose that similar results would have been observed in the medial vestibular nucleus since both thyroid hormone receptors (TRα and TRβ) are present in all vestibular nuclei.
Dutheil, S., Watabe, I., Sadlaoud, K., Tonetto, A., and Tighilet, B. (2016). BDNF signaling promotes vestibular compensation by increasing neurogenesis and remodeling the expression of potassium-chloride cotransporter KCC2 and GABAA receptor in the vestibular nuclei. J. Neurosci. 36, 6199–6212.
Lacour, M., and Tighilet, B. (2010). Plastic events in the vestibular nuclei during vestibular compensation: The brain orchestration of a “deafferentation” code. Restor. Neurol. Neurosci. 28, 19–35.
Llinas, R., and Walton, K. (1979). Vestibular compensation: A distributed property of central nerve system in Asamura H, Wilson VJ (ed) Integration in the nervous system. In A Symposium in Honor DPCL Loyd and R Lorente de No. Igaku-Shoin, Tokyo New York, p.
Luyten, W.H., Sharp, F.R., and Ryan, A.F. (1986). Regional differences of brain glucose metabolic compensation after unilateral labyrinthectomy in rats: a [14C]2-deoxyglucose study. Brain Res. 373, 68–80.
Maeda, M. (1988). Mechanisms of vestibular compensation in the unilateral labyrinthectomized cat. Prog. Brain Res. 76, 385–394.
Marouane, E., Rastoldo, G., El Mahmoudi, N., Péricat, D., Chabbert, C., Artzner, V., and Tighilet, B. (2020). Identification of New Biomarkers of Posturo-Locomotor Instability in a Rodent Model of Vestibular Pathology. Front. Neurol. 11.
McCall, A.A., Miller, D.M., and Yates, B.J. (2017). Descending Influences on Vestibulospinal and Vestibulosympathetic Reflexes. Front. Neurol. 8.
Murray, A.J., Croce, K., Belton, T., Akay, T., and Jessell, T.M. (2018). Balance Control Mediated by Vestibular Circuits Directing Limb Extension or Antagonist Muscle Co-activation. Cell Rep. 22, 1325–1338.
Péricat, D., Farina, A., Agavnian-Couquiaud, E., Chabbert, C., and Tighilet, B. (2017). Complete and irreversible unilateral vestibular loss: A novel rat model of vestibular pathology. J. Neurosci. Methods 283, 83–91.
Rastoldo, G., Marouane, E., El Mahmoudi, N., Péricat, D., Bourdet, A., Timon-David, E., Dumas, O., Chabbert, C., and Tighilet, B. (2020). Quantitative Evaluation of a New Posturo-Locomotor Phenotype in a Rodent Model of Acute Unilateral Vestibulopathy. Front. Neurol. 11.
- As the authors found the most obvious effect during the first 3 days, it may be more important to show the difference of cell differentiation in D3 rather than D30 in figure 4. The result should be added and the D30 results could be further used for analysis and discussion on its long-term effects.
The formation of new neurons sufficiently mature to integrate into a pre-existing neuronal network is a long process. We cannot observe cell differentiation in only 3 days. However, we know that 3 days after UVN a peak of cell proliferation (undifferentiated BrdU+ cells) occurs in the vestibular nuclei. A few weeks later these same cells will differentiate into neuronal cells as we demonstrated 1 month after UVN in Figure 4.
- In Figure 5 a, e, and f, the number of Iba1+ cells are less in the T4 group while it shows a higher IF intensity of KCC in D3, which seems to conflict with the original hypothesis that the gliosis of microglia decreases the KCC expression and thus facilitate the vestibular compensation. It may be better to include more discussion.
We fully agree with the reviewer's comment. The reviewer's remark is relevant. This has been taken into consideration and we have corrected this part of the discussion.
It could be better to offer detailed information on the vestibular recovery procedure measured by the weight difference applied on the left paws. The tendency of recovery is obscure here now, as the weight applied on the left paws tends to keep increasing.
Recent studies by our group have shown that Weight distribution is a robust postural stability biomarker that reflects the muscle tone applied by each leg of the animal (Facchini et al., 2021; Marouane et al., 2020, 2021; Tighilet et al., 2017). Indeed, one week after vestibular damage, the animals adopted a compensatory strategy to maintain postural stability with a shift of the body weight on the ipsilesional paws (i.e left paws) from the 7th to the 35th day (Facchini et al., 2021) and beyond (result not published). This weight asymmetry is more important in the hind paws which are essential for posture. We recently explored the functional mechanisms responsible for this postural imbalance by performing electrophysiological recordings of primary somatosensory cortex (S1) following UVN (Facchini et al., 2021). We observed, an expansion of cutaneous Receptive Fields (RFs) only on the ventral surface of the hind paws and this expansion was even greater for the ipsilesional paw at 35 days. This expansion of the RFs degraded the spatial grain of the cortical maps representing both the ipsilesional and contralesional paw. This result highlights poorer discrimination of the left hind paw. This is probably a reason why the animal applies more weight on it in order to feel it better and compensate for this alteration in discrimination.In the present study, UVN-T4 animals did not shift significantly their weight to the left even if it seems that there is an increase as reported the reviewer between day 7 and day 30. Indeed, the majority of UVN-T4 animals (as shown with the dots on Figure 1c) at D35 do not shift their weight to the left.
Interestingly, a sensorimotor rehabilitation in UVN rats that accelerates vestibular compensation also abolish the compensatory strategy to shift the body weight to the left (Marouane et al., 2021). Thus, it would seem that rats subjected to rehabilitation or pharmacological therapy (with L-T4) do not need to solicit this strategy to compensate their postural balance.
Facchini, J., Rastoldo, G., Xerri, C., Péricat, D., El Ahmadi, A., Tighilet, B., and Zennou-Azogui, Y. (2021). Unilateral vestibular neurectomy induces a remodeling of somatosensory cortical maps. Prog. Neurobiol. 205, 102119.
Marouane, E., El Mahmoudi, N., Rastoldo, G., Péricat, D., Watabe, I., Lapôtre, A., Tonetto, A., Xavier, F., Dumas, O., Chabbert, C., et al. (2021). Sensorimotor Rehabilitation Promotes Vestibular Compensation in a Rodent Model of Acute Peripheral Vestibulopathy by Promoting Microgliogenesis in the Deafferented Vestibular Nuclei. Cells 10, 3377.
Tighilet, B., Péricat, D., Frelat, A., Cazals, Y., Rastoldo, G., Boyer, F., Dumas, O., and Chabbert, C. (2017). Adjustment of the dynamic weight distribution as a sensitive parameter for diagnosis of postural alteration in a rodent model of vestibular deficit. PloS One 12, e0187472.
- Similar to Question 1 mentioned above, T4 could increase the metabolic activity of the whole body. Is there a systematic effect on behavioral scores with the administration of T4 30 minutes before behavioral tests? (Especially in open field test) It may be better to partially exclude the influence by analyzing the difference among UVN-T4, UVN-NaCl, SHAM-T4, and SHAM-NaCl groups.
We believe that intraperitoneal injections of L-T4 30 min before the behavioral tests have a systemic effect. In view of the broad spectrum of actions of thyroid hormones, we preferred to favor a global effect of the treatment rather than a localized action only at the level of the vestibular nuclei (with an injection directly into the vestibular nuclei for example). The choice of the administration route also allows us to be closer to the clinic than an invasive brain stem injection. It would have been interesting to test the T4 injections on a control animal without UVN (SHAM-T4) but this is not the purpose of our study. By choosing a global action of the low dose treatment, we increase the potential targets of L-T4 on different tissues that could facilitate vestibular compensation without triggering hypothyroidism or thyroid disorders.
- It must be noted that Supplement Figure 1. A should be corrected since the IF picture of TR and SOX2 in UVN-NaCl and UVN-T4 group at pre-operation are exactly the same one. Although the base level of 2 groups is supposed to be the same, it’s not appropriate to use the same picture.
We have corrected the IF picture of TR and SOX2 in UVN-T4 group as suggested by the reviewer. We also decide to put Supplement Figure 1 as a main Figure (now Figure 4) in the revised manuscript.
- More illustration may be needed in the result description regarding to the meaning and importance of the co-localization of Sox2 and TRa in Figure 3
According to the reviewer comment we added some explanations in the result section concerning the co-localization of Sox2 and TRα.

Reviewer 2 Report
The aim of this study is to determine if short-term treatment with L-thyroxine following vestibular neurectomy in adult rats affects the time course of vestibular recovery in the deafferented VN.
The description of the methodology is very clear, particularly the vestibular assessment (qualitative assessment of the vestibular syndrome, weight distribution, open field test, support surface), serial cell count of the vestibular nuclei and quantification of KCC2.
Figures 1b and 1c show fast recovery in day 1 to 7 for vestibular score and support surface, however no differences are found after dat 14. Could you elaborate an explanation for this? Since at the end, no differences seem to be in the outcome.
Regarding locomotion, Figure 2a and 2c only show differences on day 1-3. Could you explain this? However mean velocity and high accelerations (fig 2d and 2e) showed differences from d1 to d30, which support recovery of VOR. Could your set up measure VOR in the rat model?
Figure 4c shows quantitative assessment of the effect of UVN and L-T4 478 treatment on the number of BrdU+ cells in the deafferented MVN of UVN-NaCl (red, n=4/delay) 479 and UVN-T4 (blue, n=4/delay) groups. Here, the authors show an early increase in cell proliferation in L-T4 treated animals that is not observed in the saline-treated group; however, this is a transient effect and no differences are reported at day 30. Could you explain these findings.
Figure 5 supports the microglial reaction in in the deafferented medial vestibular nucleus after UVN.
The authors extent findings on previous work and analyzed more specifically the posturo-locomotor syndrome and demonstrated that L-T4 treatment promoted a faster recovery of postural and locomotor functions in UVN. The reported improvements in walking speed and acceleration in L-T4 treated rats should be confirmed in a randomized clinical trial.
In the discussion section, differences between the animal model of UVN and acute unilateral vestibulopathy should be highlighted since in clinical practice only occurs a partial loss of vestibular function and the outcome of L-T4 treatment could greatly differ from the rat model.
Author Response
Comments and Suggestions for Authors: Reviewer 2
The aim of this study is to determine if short-term treatment with L-thyroxine following vestibular neurectomy in adult rats affects the time course of vestibular recovery in the deafferented VN.
The description of the methodology is very clear, particularly the vestibular assessment (qualitative assessment of the vestibular syndrome, weight distribution, open field test, support surface), serial cell count of the vestibular nuclei and quantification of KCC2.
Figures 1b and 1c show fast recovery in day 1 to 7 for vestibular score and support surface, however no differences are found after dat 14. Could you elaborate an explanation for this? Since at the end, no differences seem to be in the outcome.
Figure 1b illustrates the vestibular syndrome with different behaviors that we observe in the animal once the vestibular nerve injury is performed. The resulting vestibular syndrome is divided into 2 phases: an acute phase of 1 to 7 days where the symptoms are at their peak and a compensated phase of 10 to 30 days in which the symptoms have mostly compensated. The score never returns to 0 as in pre-operation because our analysis is fine enough to highlight certain symptoms that persist, such as head tilt, which is very slight but still present. Thus, when we compare the UVN-NaCl group and the UVN-T4 group, the differences are more important during the acute phase since the vestibular disorder is the most important in this acute time window. In the compensated phase, when almost all the symptoms are no longer present, we do not show any significant difference. The L-T4 treatment will accelerate the compensation of the major deficits of the animal during the acute phase (which also corresponds to the treatment period). Within this time window, mechanisms crucial for compensation are expressed in the deafferented VN and vestibular symptoms are at their peak (Tighilet and Chabbert, 2019, for review). It is therefore logical to observe an effect of the treatment in this window which is considered both as a sensitive window of intense plasticity but also as a therapeutic window where the actions of pharmacology or rehabilitation will be the most effective.
The same logic applies for Figure 1c except that in this case the support surface of the animals after UVN returns to pre-operative values from D14. The compensation of the support surface of the UVN animals is complete and no deficit persist.
Regarding locomotion, Figure 2a and 2c only show differences on day 1-3. Could you explain this? However mean velocity and high accelerations (fig 2d and 2e) showed differences from d1 to d30, which support recovery of VOR. Could your set up measure VOR in the rat model?
Locomotion involving other spinal networks and brain regions in addition to vestibulo-spinal tracts compensates faster than the other analyzed components. Like the previous remark with vestibular score and support surface, the major deficits in locomotion occur during the first 3 days after UVN, which may explain why the differences are significant only during this period.
In the laboratory we do not have the necessary equipment to measure the VOR in rats, but the reviewer's remark is very interesting in relation to the velocity and high acceleration of UVN animals. However, in collaboration with other research team expert in VOR quantification (Beraneck group) we have recently published an article on the recovery of canal- and otolith-dependent vestibulo-ocular reflexes following UVN in mice (Cassel et al., 2019). In this study, we show that RVO is significantly impaired the mice UVN model.
Cassel, R., Bordiga, P., Carcaud, J., Simon, F., Beraneck, M., Le Gall, A., Benoit, A., Bouet, V., Philoxene, B., Besnard, S., et al. (2019). Morphological and functional correlates of vestibular synaptic deafferentation and repair in a mouse model of acute-onset vertigo. Dis. Model. Mech. 12, dmm039115.
Figure 4c shows quantitative assessment of the effect of UVN and L-T4 478 treatment on the number of BrdU+ cells in the deafferented MVN of UVN-NaCl (red, n=4/delay) 479 and UVN-T4 (blue, n=4/delay) groups. Here, the authors show an early increase in cell proliferation in L-T4 treated animals that is not observed in the saline-treated group; however, this is a transient effect and no differences are reported at day 30. Could you explain these findings.
Indeed, given the important cell proliferation observed at 3 days post-UVN in UVN-T4 animals, we would have expected a higher survival rate at 30 days. Neurogenesis is a plasticity process that requires a lot of energy, and this process starts when the vestibular lesion is brutal as in our UVN model. Milder vestibular lesions (TTX, TTK, LU) induce plasticity processes but not neurogenesis (Dutheil et al., 2011). Our interpretation of this finding is that we provide an energy booster (T4) during this critical window of plasticity. This treatment could act by restoring metabolism and excitability which are crucial parameters for vestibular compensation. In this situation of pharmacological supply of T4, the system does not solicit neurogenesis because it is an energy-consuming phenomenon and the compensatory mechanisms are supported by the T4 treatment.. We show in this study a predominantly microglial cell differentiation at the expense of new neurons at D30. These results are similar to those of another study in the laboratory which also demonstrates a reduction in neurogenesis in favor of microglial differentiation in animals submitted to a sensory-motor rehabilitation protocol (Marouane et al., 2021). The prioritization of microglial differentiation to the detriment of neurogenesis certainly has a beneficial role for the system (modulation of excitability, synaptic strenght....).
Figure 5 supports the microglial reaction in in the deafferented medial vestibular nucleus after UVN.
The authors extent findings on previous work and analyzed more specifically the posturo-locomotor syndrome and demonstrated that L-T4 treatment promoted a faster recovery of postural and locomotor functions in UVN. The reported improvements in walking speed and acceleration in L-T4 treated rats should be confirmed in a randomized clinical trial.
As suggested by the reviewer, we are currently working at the design of a randomized clinical trial aiming at validating in human the benefits of the T4 administration on the vestibular syndrome. The study will be performed in a patient population with pronounced Menière disease (repetitive crisis) in a ENT service of a French University hospital, as soon as we succeed in raising the required funding. T4 or placebo will be administered in patients and benefits on the vertigo crisis will be monitored using new clinical endpoints based on quantifiable, posturo-locomotor parameters, which we expect will be not only useful for future clinical trials for Menière disease and but also for other vestibular disorders. As this project is still in a stage of fund raising, we do not want to communicate on it in present paper.
In the discussion section, differences between the animal model of UVN and acute unilateral vestibulopathy should be highlighted since in clinical practice only occurs a partial loss of vestibular function and the outcome of L-T4 treatment could greatly differ from the rat model.
As suggested by the reviewer, it is important to point out the differences between the UVN animal model and other models of unilateral vestibulopathies, in order to anticipate the possible clinical relevance of the use of T4 as antivertigo drug. To follow his recommendations, and directly address this issue, we added a new paragraph at the end of the Discussion in a subsection on clinical relevance.

Round 2
Reviewer 1 Report
The authors explain the reasons for details of the study design in this paper based on previous work, believing that the necessity to add more animal experiments is limited. It’s acceptable according to the reference articles. And the revised edition has improved other problems concerning result description or discussion, the confusing part has also been corrected. Considering the purpose of this study, I don’t have more questions. But it would be better to make the ‘Clinical Relevance’ part more concise.
Author Response
To follow the reviewer's recommendation, we have modified the clinical relevance section in the discussion.